# Short-term power prediction of photovoltaic power stations based on Kepler optimization algorithm and VMD-CNN-LSTM model

**Jiangli Yu, Gaoyi Liang, Lei Wang**🆔\*, **Huiyuan He, Yuxin Liu, Qi Liu, Xinjie Cui, Hao Wang**

College of Electrical Engineering, Hebei University of Architecture, Zhangjiakou City, Hebei Province, China

\* wl2030@hebiace.edu.cn

## Abstract

This study focuses on the short-term power prediction of photovoltaic power stations, aiming to address the intermittent and fluctuating problems of photovoltaic power generation, in order to improve the prediction accuracy and ensure the stable operation of the power system. Innovatively introduce the Kepler algorithm into this field, deeply analyze historical data, and mine the nonlinear relationships among various factors to lay a solid data foundation for subsequent predictions. The VMD-CNN-LSTM combined model is constructed. It is a model combining variational mode decomposition (VMD), convolutional neural network (CNN) and long short term memory network (LSTM), VMD adaptively decomposes the original power sequence based on frequency characteristics to reduce data complexity. CNN accurately extracts spatial features from the decomposed modal components; LSTM leverages its expertise in processing time series data to capture the dynamic trends of power changes, and the three work in synergy. Meanwhile, the Kepler optimization algorithm (KOA) is deeply integrated with this model to optimize the entire process of the model from data preprocessing to result correction. Verified by examples, compared with the traditional prediction model, the proposed method has significant optimization in evaluation indicators such as root mean square error and mean absolute error, which strongly proves its effectiveness and superiority. It provides an innovative idea and reliable method for the short-term power prediction of photovoltaic power stations and is of great significance for promoting the grid connection of photovoltaic power generation and the optimization of the power system.

## I. Introduction

With the increasing global demand for clean energy, photovoltaic power generation, as a sustainable and pollution-free energy utilization method, has been continuously

**Data availability statement:** All relevant data are within the paper and its Supporting information files.

**Funding:** This research was supported by the Basic Research Foundation of Higher Education Institutions of Hebei Province (Grant No: 2024QNJS09). This research was also funded by the Hebei Provincial Sports Science and Technology Research Project (Project No: 2025TW10). The corresponding author is the project leader of the two projects, and the first author is the third participant of the two projects. The funders played no role in the research design, data collection and analysis, publication decisions or manuscript preparation.

**Competing interests:** The authors have declared that no competing interests exist.

rising in the proportion of the energy structure. According to the International Energy Agency (IEA), over the past decade, the global installed capacity of photovoltaic power generation has grown at an average annual rate of over 20%. As of 2023, the cumulative installed capacity has exceeded 1TW. In China, the photovoltaic industry has developed rapidly. In 2023, the cumulative installed capacity of photovoltaic power generation across the country reached 486 million kilowatts, accounting for 18.5% of the total installed capacity of power generation, becoming an important force in the energy transition.

However, the output power of photovoltaic power generation has significant intermittency and volatility. This is mainly attributed to its high dependence on meteorological conditions such as light intensity, temperature and cloud cover, as well as the diurnal and seasonal variation patterns of solar radiation [4]. For instance, during the midday on a sunny day when there is sufficient sunlight, the power output of photovoltaic power stations can reach its peak. However, on cloudy days, rainy days or in the early morning and late evening, the power will drop significantly. This kind of power fluctuation brings many challenges to the stable operation of the power system, such as grid voltage fluctuations, difficulty in frequency regulation, and in severe cases, it may even cause grid failures.

Accurate short-term power prediction of photovoltaic power stations is of great significance for the optimal dispatching of the power system, energy management and the stable operation of the power market. On the one hand, for power grid dispatching, knowing the power generation capacity of photovoltaic power stations in advance can rationally arrange the power generation plan of conventional energy, effectively reduce the phenomenon of abandoned light, and ensure the balance of power supply and demand. According to the statistics of State Grid, accurate power prediction can reduce the curtailment rate of light by 10% to 15% and improve energy utilization efficiency. On the other hand, at the energy management level, it is conducive to the coordinated control of distributed energy systems, optimizing the charging and discharging strategies of energy storage systems, and reducing the operating costs of the systems. In the electricity market environment, it can provide decision-making basis for market participants, assist in formulating reasonable electricity prices and trading strategies, and promote the healthy development of the electricity market.

Under this background, it has become an urgent task to study efficient and accurate short-term power prediction methods for photovoltaic power stations. Although traditional prediction methods such as physical models and statistical models have achieved certain results, when dealing with complex and changeable meteorological conditions and the characteristics of photovoltaic power stations, there are limitations in terms of accuracy and adaptability. With the vigorous development of artificial intelligence technology, deep learning models have demonstrated powerful nonlinear fitting and feature extraction capabilities, opening up a new path for photovoltaic power prediction.

The gravitational search algorithm (GSA), proposed by Iranian scholar Esmat Rashedi and colleagues in 2009, is a meta-heuristic optimization algorithm based on physical principles. Inspired by Newton's law of universal gravitation and the second

law of motion, GSA solves optimization problems by simulating the gravitational interactions and movement patterns of objects in the universe.

Mohamed Abdel-Basset et al. published in the journal Knowledge-Based Systems in May 2023 that, inspired by Kepler's laws of planetary motion, they simulated the orbital motion of planets for global optimization. Compared with genetic algorithm (GA), particle swarm optimization (PSO) and gravity search algorithm (GSA), KOA performs better in convergence speed and global search ability, and is especially suitable for the optimization of high-dimensional parameter Spaces. In 2024, researchers began to apply the KOA algorithm to the hyperparameter optimization of the CNN-LSTM model, such as optimizing the convolution kernel size and learning rate in load forecasting. Subsequent studies further integrated the attention mechanism to form improved versions such as KOA-CNN-LSTM-Attention [3], enhancing the model's focusing ability on key features. Model Disadvantages: Excessive model capacity: Complex network structures and optimization algorithms may lead to overfitting, especially in small sample data (such as rare disease prediction). Sensitive to noise, strict data cleaning is required (such as removing outliers and filling in missing values). The proposal of VMD In 2014, Konstantin Dragomiretskiy and Dominique Zosso proposed VMD in IEEE Transactions on Signal Processing. It aims to solve the modal aliasing problem of empirical Mode Decomposition (EMD) and is applicable to non-linear and non-stationary signal processing. The evolution of CNN-LSTM [11–16]: As an end-to-end model, CNN-LSTM was initially used for image sequence analysis (such as video prediction) and was introduced into the field of time series (such as load forecasting) in the late 20110s. The core idea is to combine the local feature extraction of CNN with the time series modeling ability of LSTM. The combination of VMD and CNN-LSTM After 2018, researchers began to use VMD as a preprocessing step to decompose the original signal in order to improve the robustness of CNN-LSTM. For instance, in 2022, Bai Wenrui et al. proposed the VMD-CNN-LSTM model [5–7] in the Journal of Sichuan University of Science and Engineering for water quality prediction in the Pearl River Basin. This is one of the early application cases of this model. Limitations are prone to fall into the local optimum trap: Both the variational problem solution of VMD and the backpropagation of CNN-LSTM may fall into local optimum, resulting in unstable model performance. Difficulty in parameter tuning: The hyperparameters such as K and α of VMD, the number of convolution kernels of CNN-LSTM, and the number of neurons of LSTM need to be jointly tuned. Manual debugging is time-consuming and dependent on experience. Automated parameter tuning tools (such as Bayesian optimization) can alleviate but cannot completely solve the problem.

This study aims to improve the accuracy of short-term power prediction for photovoltaic power stations through innovative methods, providing a basis for the stable operation and optimal dispatching of the power system. The innovation points of the research include:

1. Introduce the Kepler algorithm into the field of photovoltaic power prediction, and utilize it to analyze massive data and mine nonlinear relationships to improve the accuracy and reliability of the prediction.

2. The VMD-CNN-LSTM combined model was creatively developed. This model achieves high-precision prediction by decomposing the original power series, extracting deep spatial features and processing time series.

3. Deeply integrate the Kepler algorithm with the VMD-CNN-LSTM model to optimize each link of the model, such as data preprocessing, feature extraction and prediction result correction, in order to comprehensively improve the model performance.

VMD decomposition enhances the feature expression ability of the model. CNN-LSTM combines the capabilities of spatial feature extraction and time series modeling and is applicable to complex time series. KOA automatically optimizes hyperparameters, avoiding manual parameter adjustment and improving efficiency.

The first part of this paper introduces the Kepler algorithm and the VMD-CNN-LSTM model respectively. The second part introduces the construction of the combination of the Kepler algorithm and the VMD-CNN-LSTM model. The third part is the performance test and analysis of the algorithm. The advantages of the KOA algorithm are verified through ten

benchmark functions. Part 4 verifies the advantages of the VMD-KOA-CNN-LSTM model through experimental simulation and compares it with the other algorithms.

## Ii. Basic principles

### 2.1 Kepler algorithm

The Kepler optimization algorithm is a meta-heuristic algorithm inspired by Kepler's laws of planetary motion, which is used to predict the position and velocity of planets [1–2], as shown in Fig 1. The Kepler algorithm is an innovative method for short-term power prediction in photovoltaic power stations. It draws on the research ideas of astronomer Johannes Kepler on planetary orbits. By analyzing the relationship between historical power output and environmental factors, it establishes a mathematical model that can accurately predict future photovoltaic power generation. The core of this algorithm lies in finding a set of optimal parameters to make the predicted values as close as possible to the actual values.

   Step 1: Randomly distribute the number (population size) of N planets on the d-dimension for initialization, and the expression is as follows:

$$X_i^j = X_{i,low}^j + rand_{[0,1]} \times \left(X_{i,up}^j - X_{i,low}^j\right), \begin{cases} i = 1, 2, \ldots, N. \\ j = 1, 2, \ldots, d. \end{cases} \tag{1}$$

In the formula, $X_i$ represents the i-th planet in space (candidate solution); N represents the number of candidate solutions in the search space; d represents the dimension of the problem to be optimized; $X_{i,up}^j$ and $X_{i,low}^j$ represent the upper and lower bounds of the j-th decision variable respectively; $rand_{[0,1]}$ is used to generate random numbers from 0 to 1. The orbital eccentricity (e) of each i-th planet is initialized using the equation. The expression is as follows:

$$e_i = rand\ [0, 1], i = 1, ..., N \tag{2}$$

In the formula, $rand_{[0,1]}$ is used to generate random numbers from 0 to 1.

   Finally, initialize the orbital period T of each i-th planet. The expression is as follows

$$T_i = |r|, i = 1, ..., N \tag{3}$$

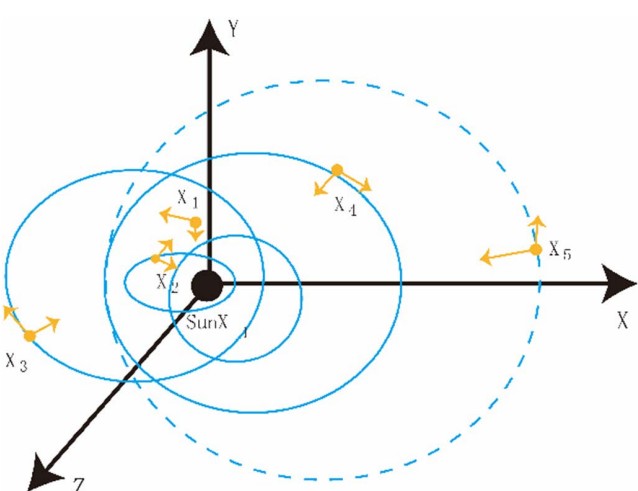

**Fig 1. Kepler optimization algorithm.**

 

In the formula, r is a number randomly generated based on the normal distribution.

Step 2: Define gravity (F). According to the law of universal gravitation, the following definitions are given for the Sun Xs and any planet $X_i$, and the expressions are as follows:

$$F_{g_i}(t) = e_i \times \mu(t) \times \frac{\overline{M}_s \times \overline{m}_i}{\overline{R}_i + \varepsilon} + r_1$$

(4)

In the formula, $\overline{M}_s$ and $\overline{m}_i$ represent the normalized values of Ms and mi, representing the mass of the sun $X_s$ and the mass of the planet $X_i$ respectively; $\varepsilon$ is a very small value; $\mu$ is the gravitational constant; $e_i$ is the eccentricity of a planet's orbit. It is a value ranging from 0 to 1, endowing KOA with random characteristics. $r_1$ is a random value ranging from 0 to 1, and the gravity varies during the optimization process. $\overline{R}_i$ represents the normalized value of $R_i$ and represents the Euclidean distance between $X_s$ and $X_i$. The expression is as follows:

$$R_i(t) = \| X_S(t) - X_i(t) \|_2 = \sqrt{\sum_{j=1}^{d} \left(X_{Sj}(t) - X_{ij}(t)\right)^2}$$

(5)

In the formula, $\|X_s(t) - X_i(t)\|_2$ represents the Euclidean distance between $X_s$ and $X_i$.

Step 3: Calculate the speed of the planet, which depends on the position of the sun. The expression is as follows:

$$V_i(t) = \begin{cases} l \times \left(2r_4\vec{X}_i - \vec{X}_b\right) + l \times \left(\vec{X}_a - \vec{X}_b\right) + (1 - R_{i-norm}(t)) \\ \times 8 \times \vec{U}_1 \times \vec{r}_5 \times \left(\vec{X}_{i,up} - \vec{X}_{i,low}\right), \text{if } R_{i-norm}(t) \leq 0.5 \\ r_4 \times 2 \times \left(\vec{X}_a - \vec{X}_i\right) + (1 - R_{i-norm}(t)) \\ \times 8 \times U_2 \times \vec{r}_5 \times \left(r_3\vec{X}_{i,up} - \vec{X}_{i,low}\right), \text{Else} \end{cases}$$

(6)

$$l = \vec{U} \times \mathcal{M} \times \mathcal{L}$$

(7)

$$\mathcal{L} = \left[\mu(t) \times (M_S + m_i) \left| \frac{2}{R_i(t) + \varepsilon} - \frac{1}{a_{i(t)} + \varepsilon} \right| \right]^{\frac{1}{2}}$$

(8)

In the formula, $V_i(t)$ represents the velocity of the object $X_i$ at time t; $r_3$ and $r_4$ are random numbers ranging from 0 to 1, and $\vec{r}_5$ is a random value vector containing values between 0 and 1. $\vec{X}_a$ and $\vec{X}_b$ represent the solutions randomly selected from the population; $\mu(t)$ represents the gravitational constant; $\varepsilon$ is a small value that prevents the error of division by zero; $a_i$ is the semi-major axis of the elliptical orbit of object i at time t. $R_i(t)$ represents the distance between the Sun $X_s$ and the target planet $X_i$ at time t; $R_i-norm(t)$ represents the normalized value of the Euclidean distance between the Sun $X_s$ and the target planet $X_i$ at time t.

Step 4: Escape from local optima within the solar system. Among them, most planets revolve counterclockwise around the Sun, and they all rotate on their own axes. However, some planets revolve around the Sun in a clockwise direction. The proposed algorithm utilizes this behavior to escape the local optimum. Simulate this behavior in the formula by modifying F in order to scan the search space accurately.

Step 5: Update the positions of the planets. By using the behavior of the planets approaching the Sun during their rotation, an equation is proposed to update the new positions of each object moving away from the Sun.

$$\vec{X}_i(t+1) = \vec{X}_i(t) + \vec{F} \times \vec{V}_i(t) + (F_{g_i}(t) + |r|) \times \vec{U} \times \left(\vec{X}_S(t) - \vec{X}_i(t)\right)$$

(10)

Step 6: Update the distance from the Sun. By adjusting the value of parameter h, when the planet approaches the Sun, KOA will focus on optimizing the exploration operation. When a planet moves away from the Sun, KOA will optimize its mining operations. The expression is as follows:

$$\vec{X}_i(t+1) = \vec{X}_i(t) + \vec{F} \times \vec{V}_i(t) + \left(F_{g_i}(t) + |r|\right) \times \vec{U} \times \left(\vec{X}_S(t) - \vec{X}_i(t)\right) \tag{11}$$

Step 7: Implement an elitist strategy to ensure the optimal position between the planet and the Sun. The expression is as follows:

$$\vec{X}_{i,new}(t+1) = \begin{cases} \vec{X}_i(t+1), \text{ iff } \left(\vec{X}_i(t+1)\right) \leq f\left(\vec{X}_i(t)\right) \\ \vec{X}_i(t), \quad \text{Else} \end{cases} \tag{12}$$

## 2.2 VMD-CNN-LSTM model

The overall design of the hybrid model is based on an innovative concept, that is, using VMD to preprocess the original time series, separate the intrinsic mode functions (IMF) of different frequency components, then extract the spatial features of these IMF with the help of CNN, and finally capture the long-term dependencies in the time series through LSTM to complete the final power prediction task, as shown in Fig 2. This structure can not only effectively deal with the nonlinear and non-stationary characteristics existing in the photovoltaic power data, but also enhance the model's adaptability to complex environmental changes.

Collect and organize the historical power output data of photovoltaic power stations and related meteorological factors, such as temperature and light intensity, as the input of the model. To ensure data quality, necessary cleaning and standardization processes must be carried out to remove outliers and missing values, and to keep all variables within the same magnitude range.

The VMD algorithm is applied to decompose the original time series data, obtaining several IMF components representing the characteristics of different frequency bands. The goal of VMD is to find a set of optimal parameters $\alpha$ (penalty factors) and $k$ (the number of IMFs), so that the decomposed IMF components can not only retain the main features of the original signal, but also reduce noise interference. The core optimization problem of VMD can be expressed as:

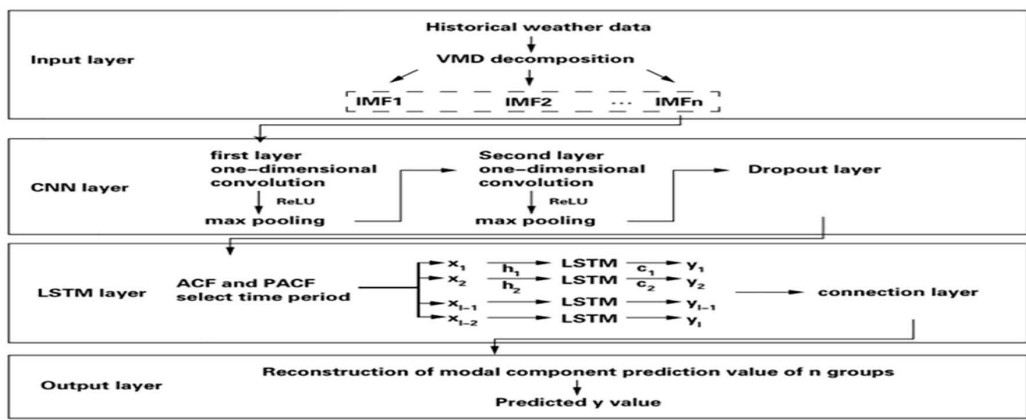

**Fig 2. VMD-CNN-LSTM model architecture.**

$$\min_{u_k, \omega_k} \left\{ \sum_{k=1}^{K} \int_0^T \left| \frac{\partial}{\partial t} \left( u_k(t) + iH\left(u_k(t)\right) \right) e^{-i\omega_k t} \right|^2 dt + \alpha \sum_{k=1}^{K} \int_0^T \left| \frac{\partial u_k(t)}{\partial t} \right| dt \right\} \tag{13}$$

Among them, $u_k(t)$ represents the ith IMF component, $\omega_k$ is center frequency, $H(\cdot)$ indicates the Hilbert transform, $\alpha$ is the parameter used to control the bandwidth penalty, and T is the length of the time window. By combining experimental verification with theoretical analysis, a set of optimal parameters can be found to ensure that the IMF components after VMD decomposition not only retain the key information of the original signal but also reduce the influence of noise to a certain extent.

Once multiple IMF components generated by VMD are obtained, the next step is to use one-dimensional or two-dimensional CNNS to extract the spatial features in these components. If the IMF is regarded as a one-dimensional time series, one-dimensional convolution is adopted; If spatial distribution is considered, two-dimensional convolution can be used. The design of CNN involves configuring multiple convolutional layers, each of which contains a certain number of filters for detecting different local patterns. The activation function is usually selected as ReLU because it can introduce nonlinearity and help improve the expressiveness of the model. The application of the pooling layer lies in reducing the feature dimension, reducing the computational complexity while also enhancing the generalization ability of the model.

For the feature maps from different IMF components, a fusion strategy was adopted, that is, they were merged into a comprehensive feature vector through cascading or other methods for further processing by the subsequent LSTM module. Let it be the feature representation of the i-th IMF component processed by CNN, then the comprehensive feature vector X can be constructed in the following way:

$$X = [x_1; x_2; ...; x_K] \tag{14}$$

The symbols here; Represent the join operation of feature vectors. Such a design not only makes full use of the information of each IMF component, but also promotes the interaction among features of different scales and enhances the ability of overall feature representation.

When the comprehensive feature vectors are ready, it enters the core part of the hybrid model – LSTM sequence modeling. At this stage, it is necessary to construct a time series suitable for LSTM input based on the comprehensive feature vectors extracted by CNN. This step may involve the sliding window technique, which combines continuous feature vectors into a series of input samples in order to better capture the continuity and dynamic changes over time. Let $X_t$ be the comprehensive eigenvector at time t, $Y_t$ represents the corresponding power outpu. Then the training set can be expressed as:

$$D = \{(X_{t-w+1:t}, Y_t) \,|t = w, w+1, ..., T\} \tag{15}$$

Here, w is the width of the sliding window and T is the total length of the time series. Build a network composed of one or more layers of LSTM units, each of which contains input gates, forget gates and output gates internally to control the information flow. The state update equation of LSTM can be written as:

$$f_t = \sigma\left(W_f \cdot [h_{t-1}, X_t] + b_f\right) \tag{16}$$

$$i_t = \sigma\left(W_i \cdot [h_{t-1}, X_t] + b_i\right) \tag{17}$$

$$o_t = \sigma\left(W_o \cdot [h_{t-1}, X_t] + b_o\right) \tag{18}$$

$$\widetilde{C}_t = \tanh\left(W_C \cdot [h_{t-1}, X_t] + b_C\right) \tag{19}$$

$$C_t = f_t \odot C_{t-1} + i_t \odot \widetilde{C}_t \tag{20}$$

$$h_t = o_t \odot \tanh\left(C_t\right) \tag{21}$$

Among them, $f_t$, $i_t$, $o_t$ are the outputs of the forget gate, input gate, and output gate respectively, $\widetilde{C}_t$ is the candidate memory state, $C_t$ is the current memory state, $h_t$ is the hidden state, W and b represent the weight matrix and bias term respectively, $\sigma$ is the activation function of $\sigma$, $\odot$ represents element-by-element multiplication. The fully connected layer is responsible for mapping the output of the LSTM to the actual predicted value $\hat{Y}_t$, such as the photovoltaic power output at a certain future time:

$$\hat{Y}_t = W_y \cdot h_t + b_y \tag{22}$$

In this way, LSTM can not only effectively learn the long-term dependencies in the time series, but also flexibly cope with time intervals of different lengths, providing strong support for prediction.

In order to ensure that the hybrid model can achieve the best effect in practical applications, the model training and optimization links cannot be ignored either. In the selection of the loss function, considering the importance of the prediction error, MSE was chosen as the measurement criterion. However, apart from MSE, MAE and other appropriate loss functions can also be adjusted according to specific needs. The loss function is defined as follows:

$$L(\theta) = \frac{1}{N}\sum_{i=1}^{N}\left(Y_i - \hat{Y}_i\right)^2 \tag{23}$$

Among them, $\theta$ represents the model parameters, N is the sample size, and $Y_i$ and $\hat{Y}_i$ are the true values and predicted values respectively. Secondly, the selection of the optimization algorithm directly affects the convergence speed and final performance of the model. Therefore, efficient optimization algorithms, such as Adam or RMSprop, were adopted with the goal of minimizing the loss function. Meanwhile, setting appropriate hyperparameters such as the learning rate and batch processing size can accelerate the convergence speed and avoid getting trapped in local optimal solutions. In addition, to prevent model overfitting, L1/L2 regularization terms have been added or techniques such as Dropout have been used. The Early Stopping method is also implemented. When the performance on the validation set no longer improves, the training is terminated in advance to maintain the optimal state of the model.

## Iii. Construction based on Kepler algorithm and VMD-CNN-LSTM model

The short-term power prediction model for photovoltaic power stations proposed in this study based on the Kepler algorithm and the VMD-CNN-LSTM model integrates multiple advanced technologies in its overall architecture, aiming to fully explore the potential patterns in photovoltaic power data and achieve high-precision prediction, as shown in Fig 3.

The data preprocessing module, as the basic link, is responsible for operations such as cleaning and normalizing the original photovoltaic power station data. Due to the influence of factors such as sensor accuracy and environmental interference on photovoltaic power data, there are often abnormal values and missing values. Data cleaning identifies and corrects or eliminates these abnormal data by setting reasonable thresholds to ensure data quality. Normalization unifies data of different dimensions to the same scale. For example, by using the Z-score normalization method, the data

is transformed into a distribution with a mean of 0 and a standard deviation of 1, which improves the efficiency of model training and provides reliable data input for subsequent modules.

The VMD module decomposes the preprocessed photovoltaic power time series and adaptively splits the complex non-stationary series into multiple intrinsic modal functions (IMFs) with different center frequencies based on the characteristics of the signal itself. For example, the original power sequence containing high-frequency noise, the influence of medium-frequency weather changes, and low-frequency seasonal trends is decomposed into relatively independent IMF components to reduce data complexity, enabling subsequent models to focus on the feature learning of each component and avoid mutual interference among different frequency components.

The CNN module automatically extracts rich local features for each IMF component obtained by VMD decomposition by taking advantage of the weight sharing and local perception characteristics of its convolutional layers. By designing convolution kernels of different scales, such as 3 × 1 and 5 × 1 convolution kernels sliding in the time dimension, the characteristic patterns such as sudden change points and periodic fluctuations of the power curve are captured. The complex time series is transformed into more representative abstract feature maps, providing the LSTM module with initially refined input rich in key information. Reduce the study burden of LSTM.

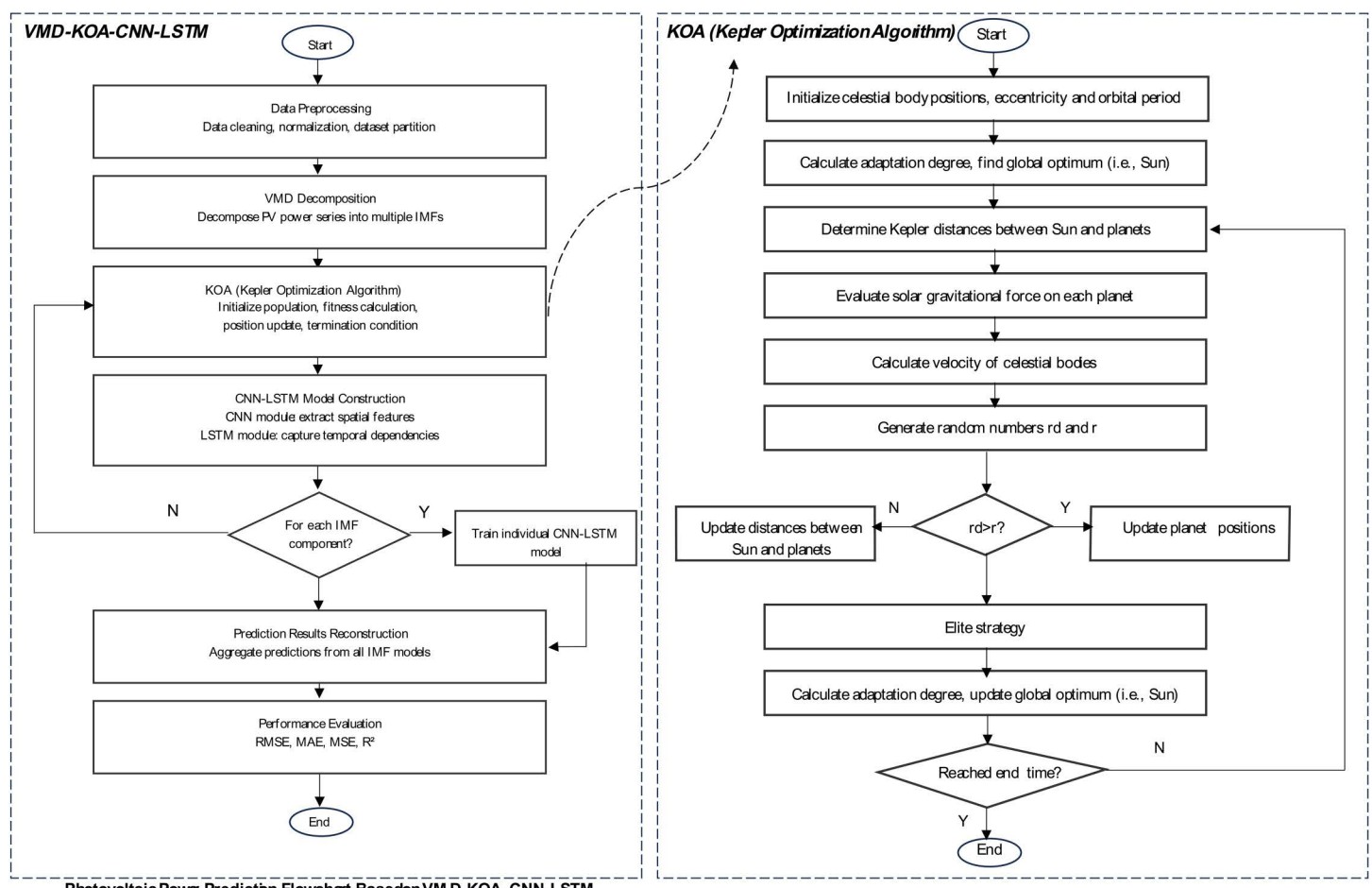

**Fig 3. VMD-KOA-CNN-LSTM model architecture.**

The LSTM module models the long-term dependencies in time series data based on the feature maps extracted by CNN and with its unique gating mechanism. Its input gate, forget gate and output gate work in coordination. When dealing with situations such as continuous overcast days causing persistently low power and periodic power changes due to seasonal changes, it can effectively integrate historical information, accurately capture the long-term trend of power changes, and dynamically adjust the transmission and retention of information at each moment based on information such as power values and meteorological conditions at past moments. Output prediction results that are in line with the actual situation.

The Kepler algorithm optimization module, as a key innovation point of the model, runs through the model training process. On the one hand, the Kepler algorithm is utilized to conduct in-depth analysis of historical power data. Based on its ability to simulate the orbits of celestial bodies and accurately identify periodic laws, periodic characteristics such as seasonality and diurnal variations of power changes are mined, providing key prior knowledge for model prediction and enhancing the adaptability of the model to power fluctuations in different periods. On the other hand, during the model training stage, the Kepler algorithm compares the model parameters to be optimized (such as the convolution kernel weights of CNN and the number of neurons in the hidden layer of LSTM) to the positions and velocities of celestial bodies. By simulating the gravitational effects and velocity updates during the planetary motion process, it quickly screens out similar patterns in massive historical data. Provide more targeted training samples for the model, accelerate the convergence of the model and improve the prediction efficiency.

Each module works closely together to form a complete process from data preprocessing, feature decomposition and extraction to sequence modeling and optimization, giving full play to their respective advantages and providing a strong guarantee for accurately predicting the short-term power of photovoltaic power stations.

The model architecture is shown in Fig 3, mainly covering the data preprocessing module, the variational mode decomposition (VMD) module, the convolutional neural network (CNN) module, the Long Short-Term Memory Network (LSTM) module, and the Kepler algorithm optimization module.

## Iv. Algorithm performance testing and analysis

### 4.1 Benchmark test function

The content of this section is to conduct experiments on KOA. KOA is selected as the experimental group, and the SSA algorithm, PSO algorithm, GWO algorithm, and WOA algorithm are selected as the control group. Ten test functions including unimodal and multimodal functions are selected for the experiments. Through the analysis of the optimal value, the worst value, The performance of these five algorithms is analyzed based on the data of the mean and standard deviation as well as the convergence curve graph. The smaller the mean value is, the stronger the algorithm's ability to avoid falling into local optimum and global search is. The smaller the standard deviation is, the stronger the robustness of the algorithm is. The worst value and the best value represent the best and worst fitness values respectively. The CEC 2005 optimization functions (F1 to F10) Each function contains the expression of the objective function as well as the corresponding S (lb), S upper limit (ub), and dimension (n_dim) [18]. Benchmark test functions are an important way to measure the effectiveness of algorithms. To evaluate the effectiveness of the Kepler algorithm, ten groups of test functions, including unimodal test functions, multimodal test functions and fixed-dimensional test functions, were selected to conduct performance tests on the Kepler algorithm in this paper. The purpose of conducting benchmark test function experiments is to evaluate the ability of different algorithms to fall into local optimum and global search during the operation process by using the optimal value, the worst value and the mean standard deviation.

### 4.2 Test results and analysis

In order to test the effectiveness of the KOA algorithm, comparative experiments were conducted with the PSO, GWO, SSA and WOA algorithms, and the common parameters of all algorithms remained consistent.

The five algorithms were run continuously and independently on the benchmark test functions F1 to F10 to obtain the average value, standard deviation, worst value and optimal value of the five algorithms. Among them, the smaller the mean value, the stronger the algorithm's ability to avoid falling into local optimum and global search. The smaller the standard deviation, the stronger the robustness of the algorithm. The worst value and the best value represent the best and worst fitness values respectively.

To visually demonstrate their properties, three-dimensional stereoscopic images of the benchmark test functions were constructed respectively, and a one-draw iterative convergence comparison curve similar to the average result in the test was taken. The figure shows the three-dimensional images and convergence curves of the benchmark test functions F1 to F10. By comparing and analyzing the convergence curves of each algorithm, it is possible to more intuitively compare the convergence of each algorithm and its ability to break out of the local space.

By analyzing the four curves of F1 in the Fig 4, it can be seen that KOA has the fastest convergence speed during the iterative process and finally achieves the expected convergence, while PSO shows the slowest convergence speed. Finally, the GWO, WOA, SSA and PSO algorithms all fail to achieve the expected convergence accuracy.

By analyzing the four curves of F2 in the Fig 5, it can be seen that KOA has the fastest convergence speed during the iterative process and finally achieves the expected convergence, while PSO shows the slowest convergence speed. Finally, the GWO, WOA, SSA and PSO algorithms all fail to achieve the expected convergence accuracy.

By analyzing the four curves of F3 in the Fig 6, it can be seen that KOA has the fastest convergence speed during the iterative process and finally achieves the expected convergence, while PSO and WOA show the slowest convergence speed. Finally, the GWO, WOA, SSA and PSO algorithms all fail to achieve the expected convergence accuracy.

By analyzing the four curves of F4 in the Fig 7, it can be seen that KOA has the fastest convergence speed during the iterative process and finally achieves the expected convergence, while PSO and WOA show the slowest convergence speed. Finally, the GWO, WOA, SSA and PSO algorithms all fail to achieve the expected convergence accuracy.

By analyzing the four curves of F5 in the Fig 8, it can be seen that KOA has the fastest convergence speed during the iterative process and finally achieves the expected convergence, while PSO shows the slowest convergence speed. Finally, the PSO algorithm fails to reach the expected convergence accuracy.

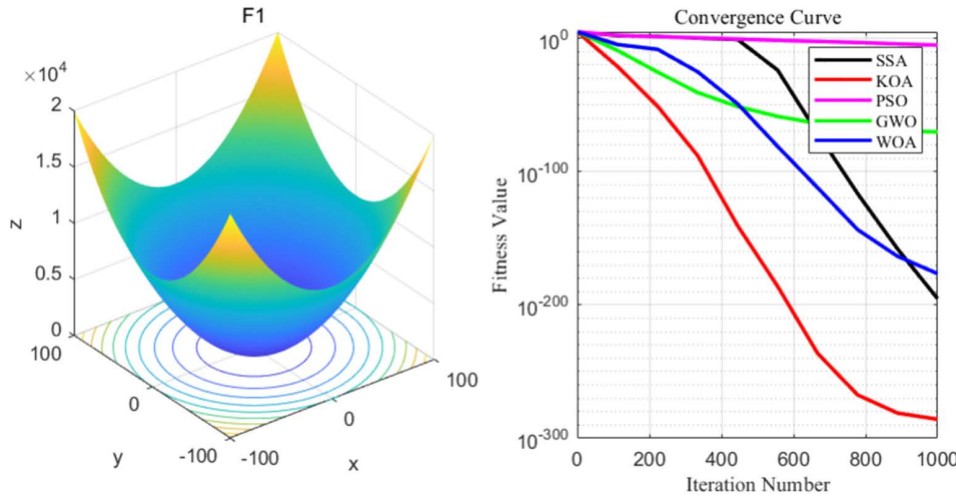

**Fig 4. F1 function and convergence curve graph.**

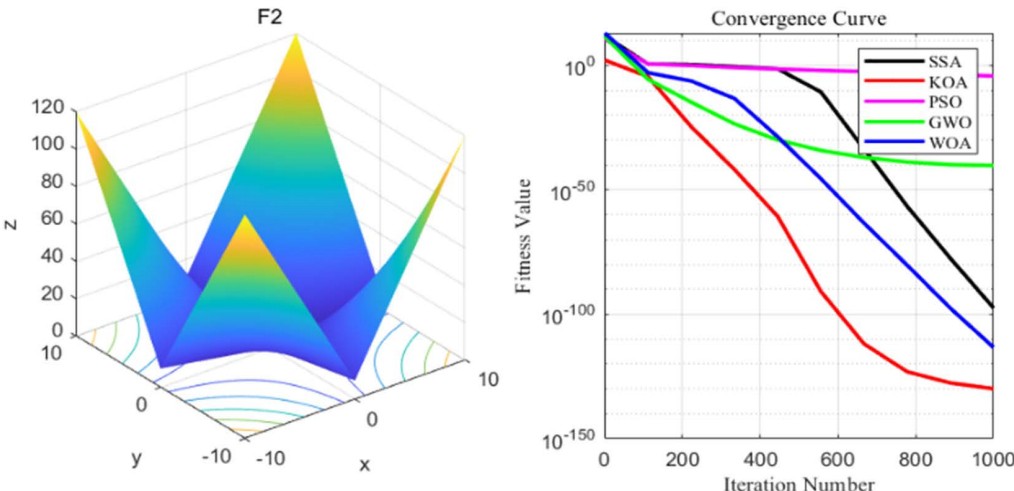

**Fig 5. F2 function and convergence curve graph.**

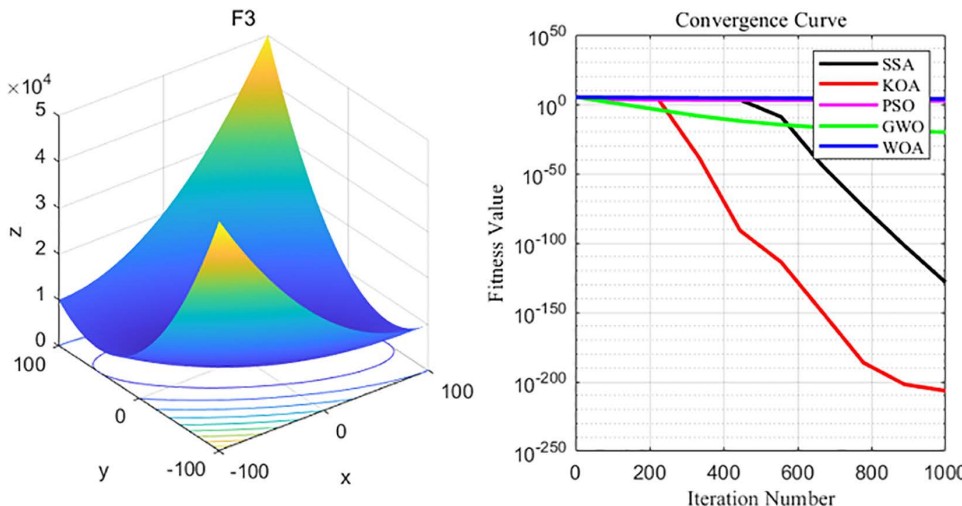

**Fig 6. F3 function and convergence curve graph.**

By analyzing the four curves of F6 in the Fig 9, it can be seen that KOA has the fastest convergence speed during the iterative process and finally achieves the expected convergence, while GWO and SSA show the slowest convergence speed. Finally, the GWO, WOA, SSA and PSO algorithms all fail to achieve the expected convergence accuracy.

By analyzing the four curves of F7 in the Fig 10, it can be seen that SSA has the fastest convergence speed during the iterative process and finally achieves the expected convergence, while PSO shows the slowest convergence speed. Finally, the GWO, WOA and PSO algorithms all fail to achieve the expected convergence accuracy.

By analyzing the four curves of F8 in the Fig 11, it can be seen that WOA has the fastest convergence speed during the iterative process and finally achieves the expected convergence, while GWO shows the slowest convergence speed. Finally, neither the GWO nor the PSO algorithm achieves the expected convergence accuracy.

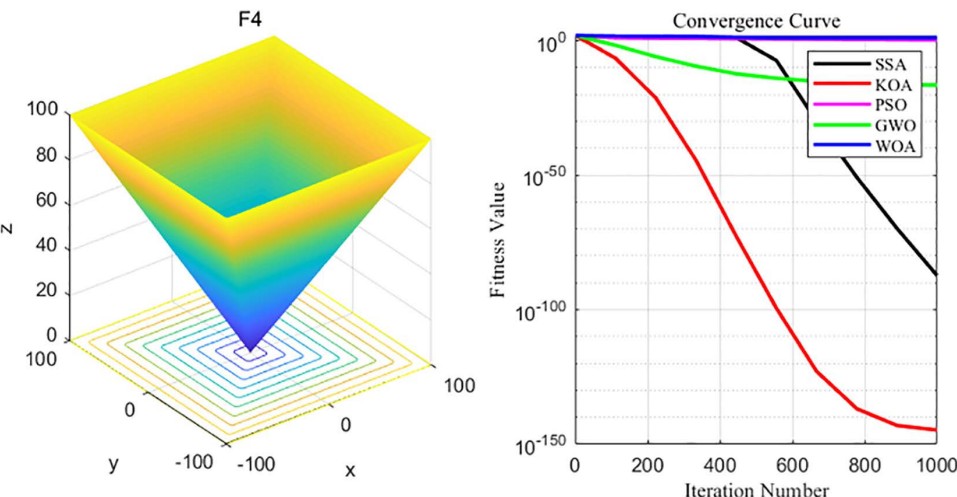

**Fig 7. F4 function and convergence curve graph.**

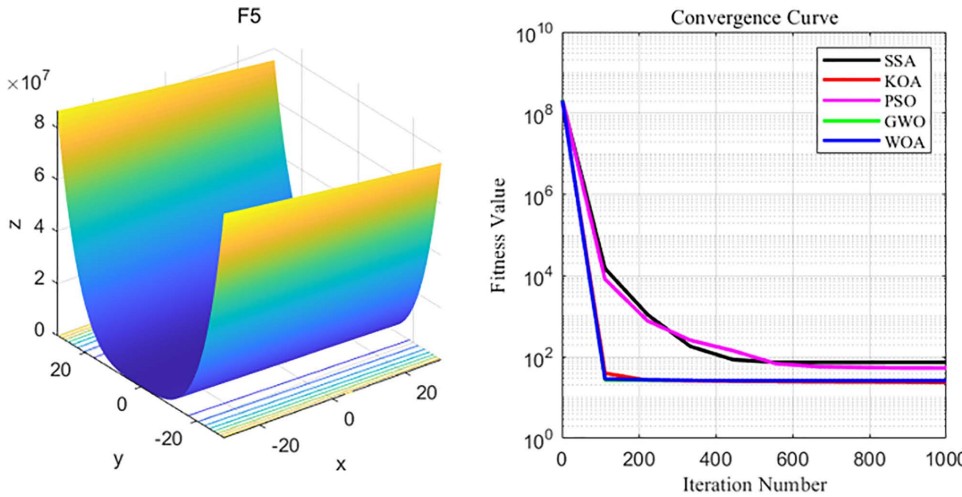

**Fig 8. F5 function and convergence curve graph.**

By analyzing the four curves of F9 in the Fig 12, it can be seen that KOA has the fastest convergence speed during the iterative process and finally achieves the expected convergence, while PSO and SSA show the slowest convergence speed. Finally, the GWO, WOA, SSA and PSO algorithms all fail to achieve the expected convergence accuracy.

By analyzing the four curves of F10 in the Fig 13, it can be seen that KOA has the fastest convergence speed during the iterative process and finally achieves the expected convergence, while PSO shows the slowest convergence speed. Finally, the GWO, WOA, SSA and PSO algorithms all fail to achieve the expected convergence accuracy.

To sum up, among the 10 benchmark test functions, compared with the MSSA, SSA, PSO and SCA algorithms, it indicates that the Kepler algorithm has certain advantages in terms of convergence speed, the number of iterations and the ability to escape the local optimum. Specifically, under multiple test functions, KOA converges faster and is more likely to converge to the theoretical optimal value.

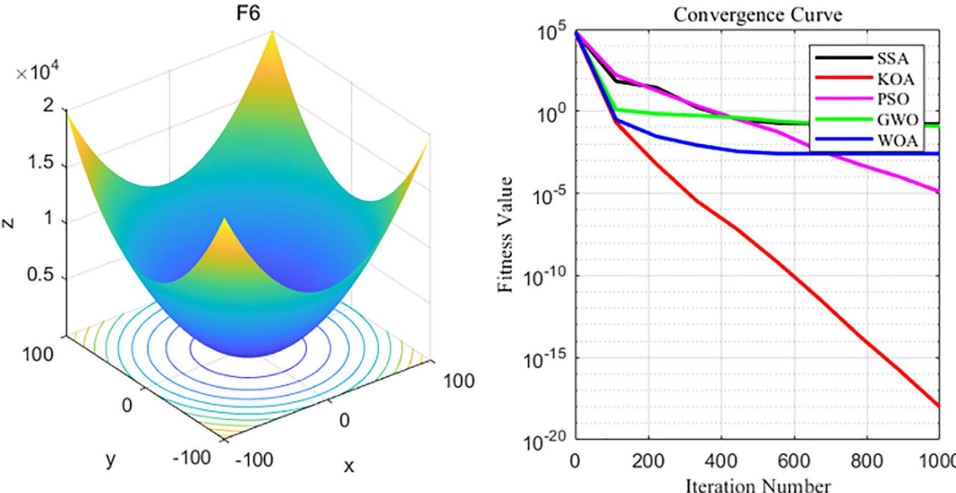

**Fig 9. F6 function and convergence curve graph.**

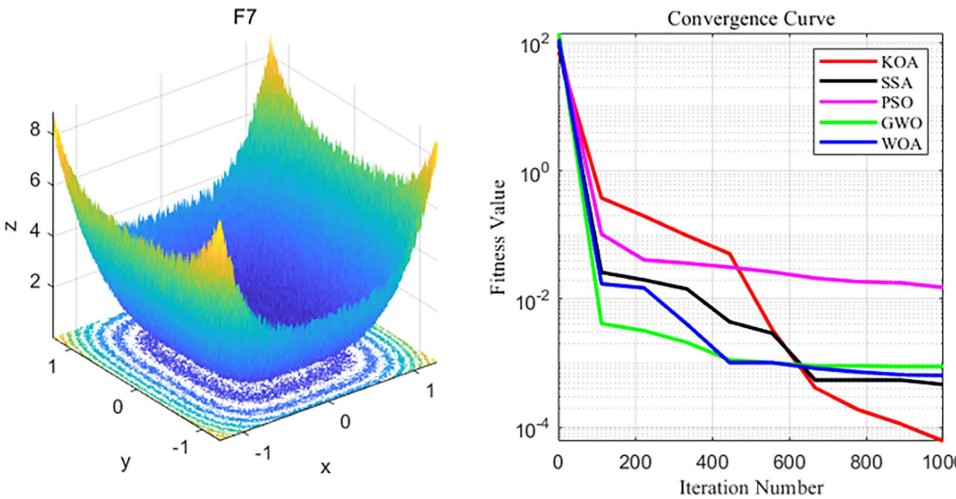

**Fig 10. F7 function and convergence curve graph.**

## V. Case verification and result analysis

### 5.1 Data sources

This study utilized data from two 50MW and 130MW photovoltaic power stations (Station site1 and Station site2) in China's State Grid Renewable Energy Power Generation Forecasting Competition [17]. The dataset covers the period from 2019 to 2020, with hourly records of power generation and weather-related data, including total solar radiation, direct normal radiation, global horizontal radiation, air temperature, atmospheric pressure, relative humidity, and the total power generation of the photovoltaic power stations. Since photovoltaic panels do not generate power at night, the midnight data is not representative for analyzing daytime power generation efficiency. By excluding this data, the dataset becomes more representative, which helps the model learn better and more accurately capture the patterns and trends of daytime photovoltaic power generation. The training set and test set were divided in an 8:2 ratio.

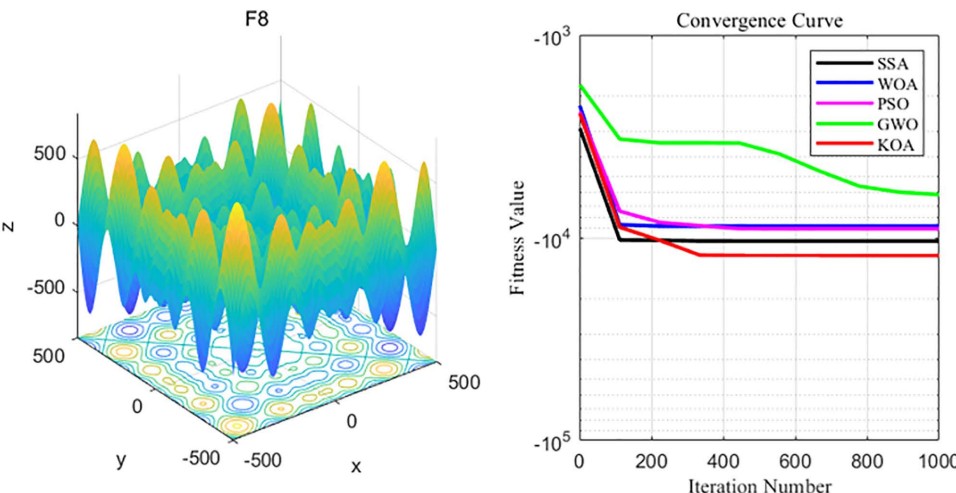

**Fig 11. F8 function and convergence curve graph.**

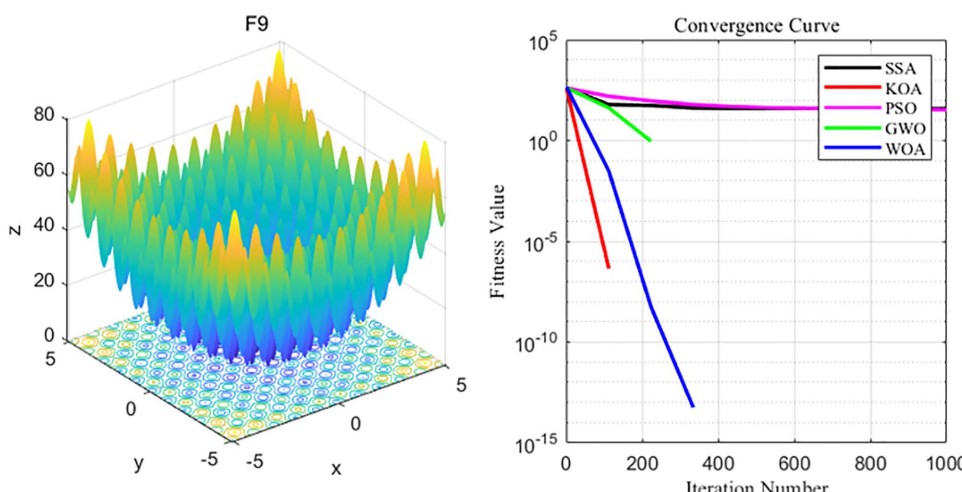

**Fig 12. F9 function and convergence curve graph.**

## 5.2 Comparison of prediction results

In order to comprehensively evaluate the prediction performance of the VMD-KOA-CNN-LSTM model constructed in this study, several models were selected for comparative analysis. Including CNN-LSTM, KOA-CNN-LSTM, VMD-LSTM [8–10], and the VMD-CNN-LSTM model that is not optimized by combining the Kepler algorithm.

On the same test set data, MAE (Mean Absolute Error): Mean absolute error, describing the degree of deviation between the predicted value and the true value. RMSE (Root Mean Squared Error): The root mean square error, which measures the average magnitude of the prediction error. R² (R-Squared): Determination coefficient, representing the fitting effect of the model, ranging from 0 to 1. The closer to 1, the better. For instance, for a set of photovoltaic power prediction data of $N$ samples, let the actual power value be $Y_i$ and the predicted power value be $\hat{Y}_i$, then the calculation formula is:

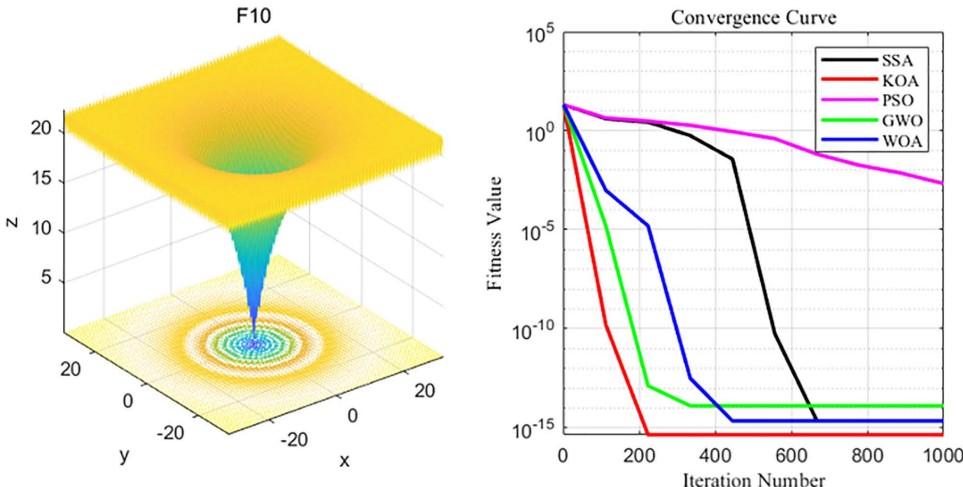

**Fig 13. F10 function and convergence curve graph.**

$$MAE = \frac{1}{N} \sum_{i=1}^{N} \left| Y_i - \hat{Y}_i \right|$$

(24)

$$RMSE = \sqrt{\frac{1}{N} \sum_{i=1}^{N} \left( Y_i - \hat{Y}_i \right)^2}$$

(25)

$$R^2 = 1 - \frac{\sum_{i=1}^{N} \left( Y_i - \hat{Y}_i \right)^2}{\sum_{i=1}^{N} \left( Y_i - \bar{Y} \right)^2}$$

(26)

As shown in Table 1. Data comparison of various models at station site1, compared to other models, this study's Kepler optimization algorithm and VMD-CNN-LSTM model show significant improvements in various metrics. In terms of root mean square error (RMSE), it is 1.89 lower than the traditional CNN-LSTM model, 0.26 lower than the VMD-LSTM model, 1.12 lower than the ordinary KOA-CNN-LSTM model, and 0.06 lower than the same model without Kepler optimization. The mean absolute error (MAE) is also superior to other models, with reductions of 1.59,0.19,0.57, and 0.08. As shown in Table 2. Data comparison of various models at station site2, compared to other models, this study's Kepler algorithm and VMD-CNN-LSTM model show significant improvements in various metrics. In terms of RMSE, it is 3.68 lower than the traditional CNN-LSTM model, 0.14 lower than the VMD-LSTM model, 2.97 lower than the ordinary KOA-CNN-LSTM model, and 0.3 lower than the same model without Kepler optimization. The MAE is also superior to other models, with reductions of 2.92,0.07,2.27, and 0.31. Particularly in the R² metric, which measures goodness of fit, it is close to 1, indicating that the model can effectively capture the trend of photovoltaic power output changes. This demonstrates that the deep integration of the Kepler algorithm with the model significantly enhances prediction accuracy, accurately captures power changes, and reduces errors. In

**Table 1. Data comparison of various models at station site1.**

| Model name | MAE | RMSE | R² |
|---|---|---|---|
| CNN-LSTM | 3.36 | 4.28 | 0.85 |
| VMD-LSTM | 1.96 | 2.65 | 0.94 |
| KOA-CNN-LSTM | 2.34 | 3.51 | 0.90 |
| VMD-CNN-LSTM (Notoptimized) | 1.85 | 2.45 | 0.95 |
| VMD-KOA-CNN-LSTM | 1.77 | 2.39 | 0.95 |

**Table 2. Data comparison of various models at station site2.**

| Model name | MAE | RMSE | R² |
|---|---|---|---|
| CNN-LSTM | 5.18 | 6.81 | 0.93 |
| VMD-LSTM | 2.33 | 3.27 | 0.98 |
| KOA-CNN-LSTM | 4.53 | 6.10 | 0.98 |
| VMD-CNN-LSTM (Notoptimized) | 2.57 | 3.43 | 0.94 |
| VMD-KOA-CNN-LSTM | 2.26 | 3.13 | 0.98 |

addition to quantitative comparisons, the prediction results can also be visualized to highlight the performance differences between models.

Figs 14 and 15 are VMD-KOA-CNN-LSTM model operation diagram and trend iteration diagram respectively (Station site1). Figs 16 and 17 are VMD-KOA-CNN-LSTM model operation diagram and trend iteration diagram respectively (Station site2).

Figs 18 and 19 present the comparison between the prediction results of the KOA-CNN-LSTM and VMD-KOA-CNN-LSTM models and the actual power curves. It is evident that while the KOA-CNN-LSTM model can capture some temporal variations, it exhibits lag and insufficient accuracy when dealing with complex meteorological conditions and long-term trends. Photovoltaic power generation is highly non-stationary and complex, with fluctuations resulting from the superposition of multiple physical processes at various time scales: from cloud shadow shifts and equipment noise at the second level, to local meteorological changes and cloud evolution over minutes to hours, to weather system activities on a daily to weekly scale, and even diurnal and seasonal changes. These fluctuations at different frequencies are intertwined in the original power signal, making it challenging for direct prediction models to accurately capture their underlying patterns and are sensitive to noise.

To address the aforementioned challenges, Variational Mode Decomposition (VMD) can adaptively decompose complex signals into a series of physically meaningful eigenmode functions (IMFs) based on the frequency characteristics of the signal. Each IMF has a well-defined central frequency, clearly corresponding to core driving factors at different time scales, such as cloud shadows, air mass movements, and day-night or seasonal cycles. This decomposition achieves signal decoupling:

(1) Eliminate interference: can identify and filter out high frequency noise (IMF1–2).

(2) Separation of prediction focus: the predictable components at different scales (short-term, medium-term and long-term) (such as local cloud change (IMF3), weather system influence (IMF4), day and night patterns and seasonal trends (IMF5–6)) are effectively separated to avoid mixing.

(3) Reduce data complexity and enhance modeling accuracy: The decomposition of sequences enhances their stability and regularity, making it possible to tailor prediction models for different frequency components (for example, using short-term response models for high-frequency components and trend capture models for low-frequency

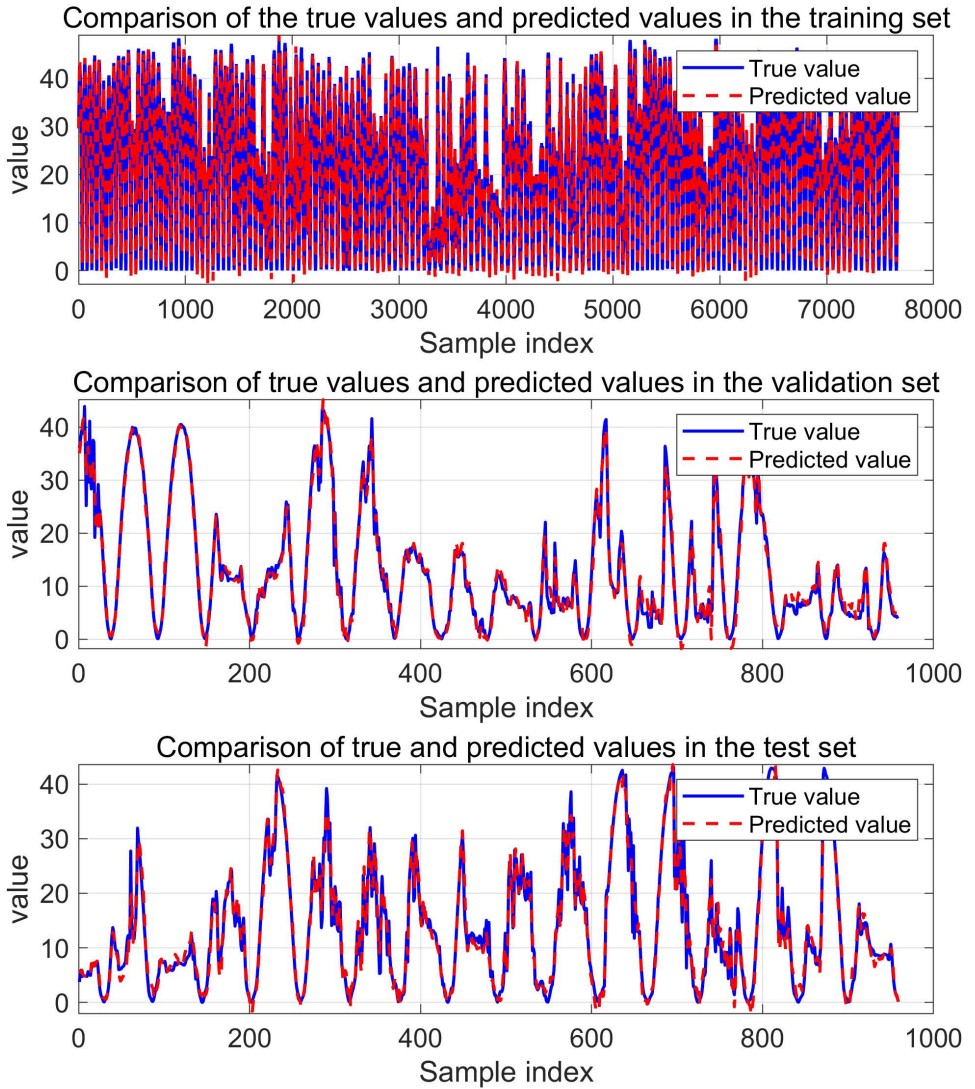

**Fig 14. The operation diagram of the VMD-KOA-CNN-LSTM model (Station site1).**

components). This approach ultimately leads to more accurate overall power predictions by combining the results from each component.

Therefore, VMD decomposition is an effective and necessary processing method to overcome the complexity of photovoltaic power data and improve its predictability.

Figs 20 and 21 show the comparison between the prediction results of VMD-CNN-LSTM (unoptimized) and VMD-KOA-CNN-LSTM models and the actual power curve. The unoptimized VMD-CNN-LSTM model has improved, can extract more features and respond more sensitively, but there are still details of deviation.

The significance and algorithm advantages of KOA optimization of CNN-LSTM hyperparameters

(1) The significance of hyperparameter optimization: In photovoltaic power prediction, the CNN-LSTM model involves several key hyperparameters, including the number and size of convolutional kernels in the CNN layer, the

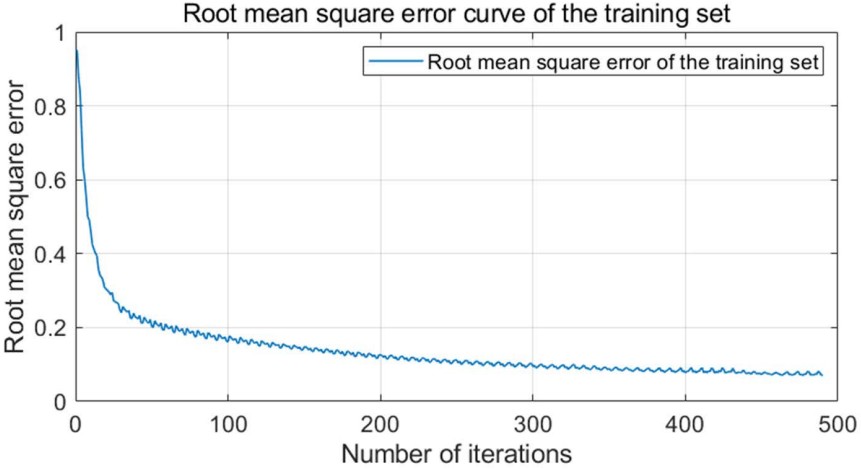

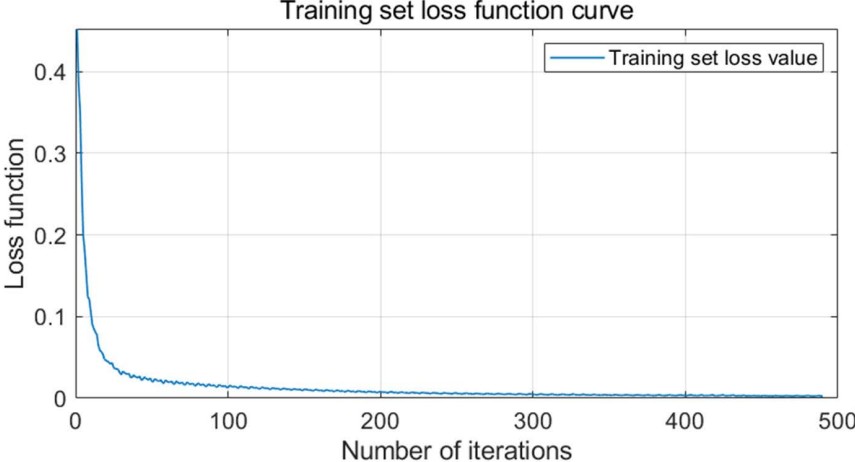

**Fig 15. Iterative Trend chart (Station site1).**

number of hidden units and layers in the LSTM layer, dropout rates, and optimizer parameters such as learning rate and batch size. Traditional manual parameter tuning or grid search is inefficient in such a high-dimensional parameter space and can easily lead to local optima. Moreover, there is a strong coupling effect among these hyperparameters. For example, when the number of CNN convolutional kernels increases, the corresponding number of LSTM hidden units and learning rate need to be adjusted accordingly; otherwise, issues like gradient disappearance or overfitting may arise. Optimizing each parameter individually cannot capture this interdependence, whereas KOA's global search mechanism can consider the synergistic effects of all parameters simultaneously.

(2) The advantages of KOA over other optimization algorithms: KOA is based on the Keplerian laws of astrophysics, and its mechanism of 'planets' orbiting the 'optimal solution sun' has a clear physical basis. This avoids the randomness in parameter settings found in other heuristic algorithms. Parameters such as gravitational coefficients and orbital radii have clear physical explanations, making them easier to understand and adjust. Compared to other

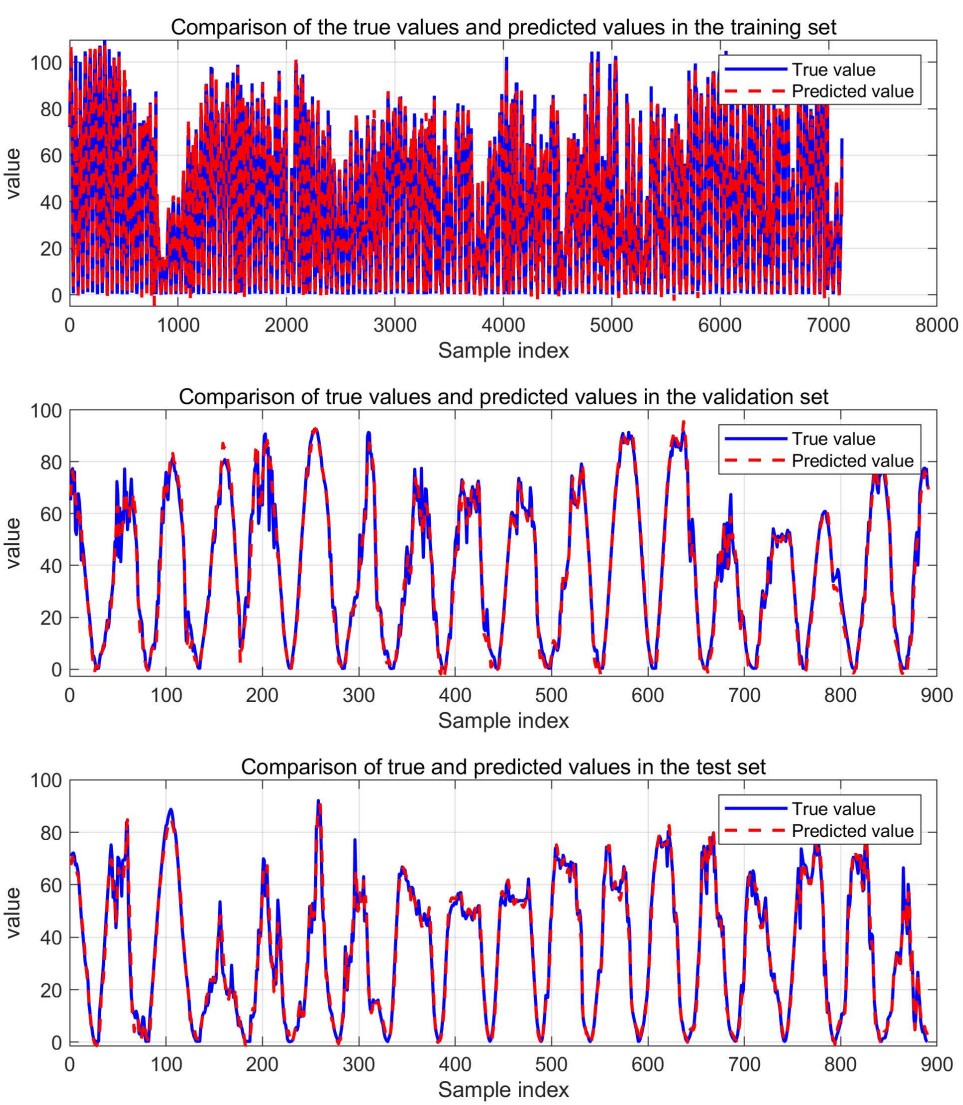

**Fig 16. The operation diagram of the VMD-KOA-CNN-LSTM model (Station site2).**

optimization algorithms, KOA converges faster and can achieve global optimization, leading to more robust parameter configurations.

The prediction curve of the research model is highly consistent with the actual situation, and the stable or fluctuating period is accurately predicted, which is almost coincident. It intuitively verifies its excellent performance in dealing with photovoltaic power problems, and provides guarantee for the stable operation and optimal scheduling of the power station.

## 5.3 Conclusion

This study focuses on the short-term power prediction problem of photovoltaic power stations and innovatively proposes a prediction method based on the Kepler optimization algorithm and the VMD-CNN-LSTM model. Through in-depth

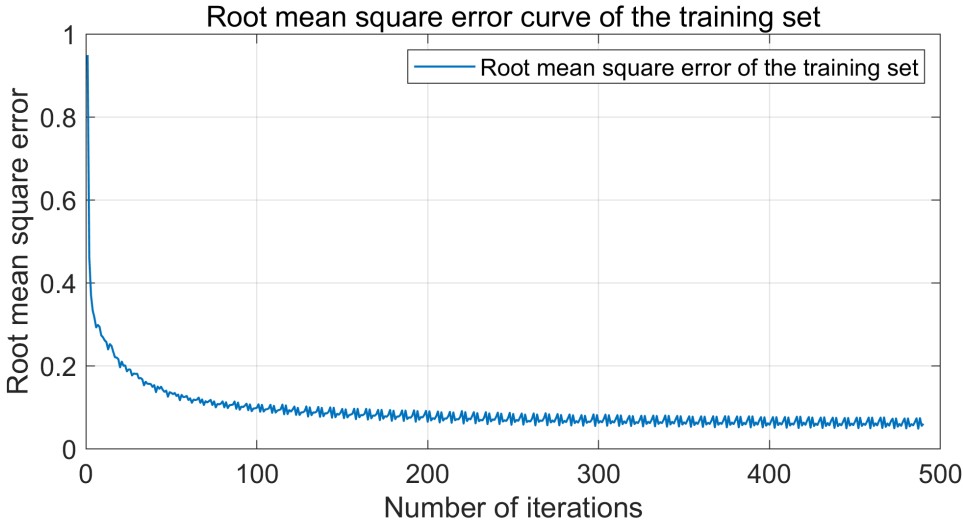

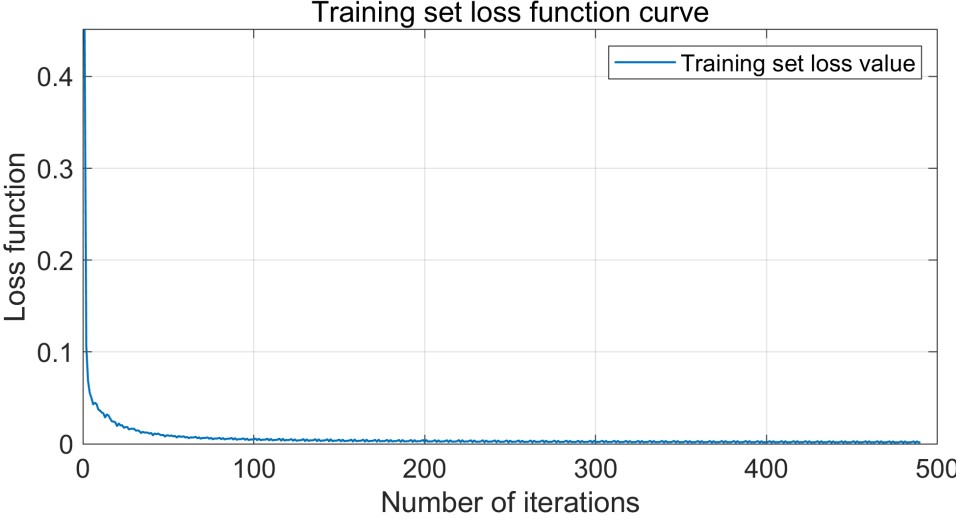

**Fig 17. Iterative Trend chart (Station site2).**

analysis and processing of the data of actual photovoltaic power stations, the validity and superiority of this model have been verified. The model has significant advantages: Firstly, the Kepler algorithm has powerful global search and feature mining capabilities, simulates celestial motion to capture nonlinear relationships, provides key support for subsequent links, prevents overfitting, and improves generalization and accuracy; Second, the components of the VMD-CNN-LSTM architecture work collaboratively. VMD decomposes the stationary components of the sequence, reduces complexity, and highlights features. CNN extracts deep features and captures short-term fluctuations. LSTM combines local and historical information to track long-term changes and achieve high-precision prediction. Compared with traditional prediction models, the model in this study has been significantly optimized in key evaluation indicators such as root mean square error and mean absolute error. It can predict the short-term power changes of photovoltaic power stations more accurately, providing reliable decision-making basis for departments such as power grid dispatching and energy management, and

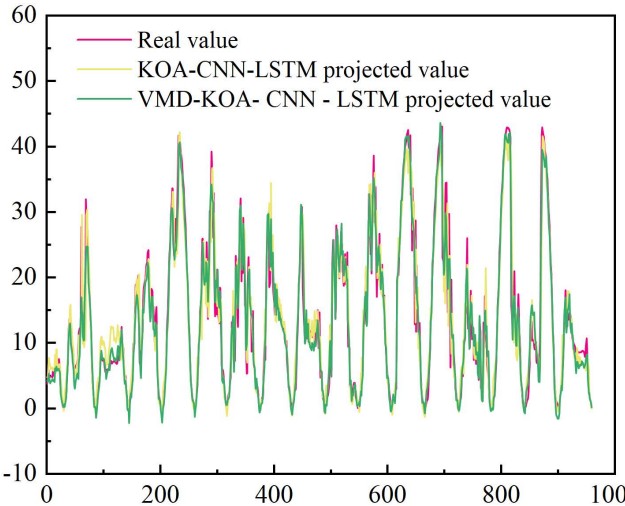

**Fig 18. KOA-CNN-LSTM and VMD-KOA-CNN-LSTM Comparison of the model prediction results with the actual power (Station site1).**

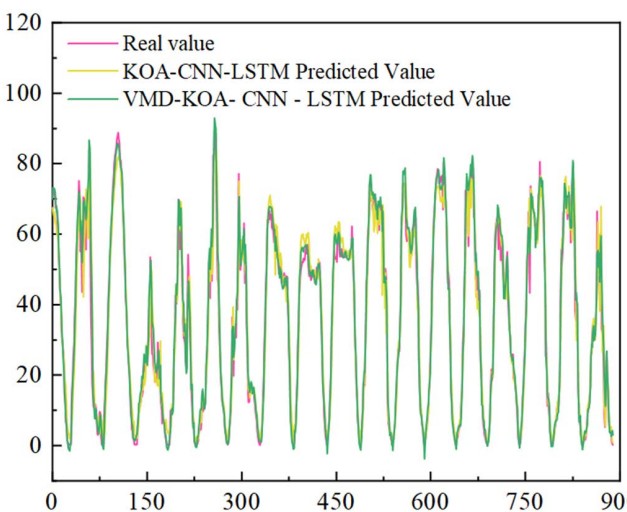

**Fig 19. KOA-CNN-LSTM and VMD-KOA-CNN-LSTM Comparison of the model prediction results with the actual power (Station site2).**

effectively ensuring the stable and economically efficient operation of the power system. It has significant practical significance for increasing the proportion of photovoltaic power generation in the energy structure and promoting the wide application of clean energy.

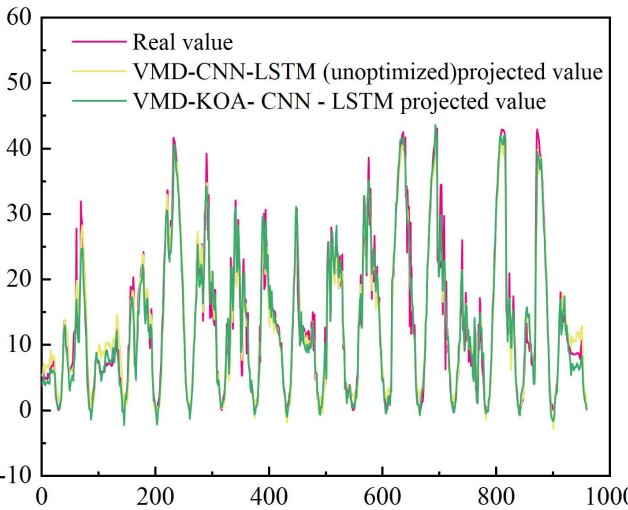

**Fig 20. VMD-CNN-LSTM (unoptimized) and VMD-KOA-CNN-LSTM Comparison of the model prediction results with the actual power (Station site1).**

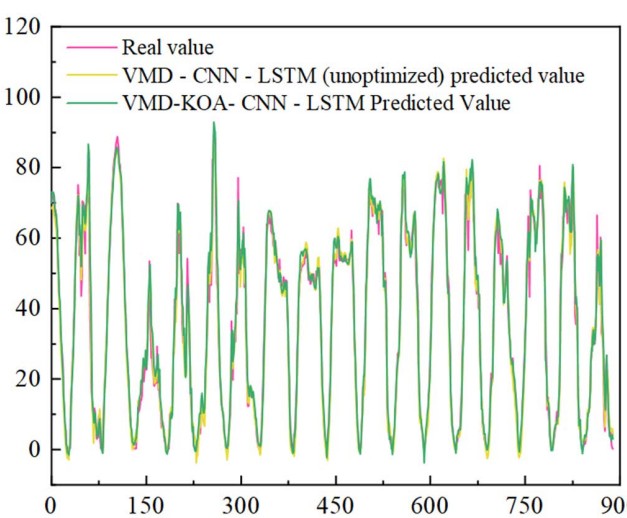

**Fig 21. VMD-CNN-LSTM (unoptimized) and VMD-KOA-CNN-LSTM Comparison of the model prediction results with the actual power (Station site2).**

## Supporting information

**S1 File. Station site1.**
(CSV)

**S2 File. Station site2.**
(CSV)

## Author contributions

**Conceptualization:** lei wang.

**Data curation:** Jiangli Yu, Huiyuan He.

**Formal analysis:** Yuxin Liu.

**Investigation:** Qi Liu, Hao Wang.

**Methodology:** Xinjie Cui.

**Writing – original draft:** Gaoyi Liang.

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
