## [Decision Letter · Decision Letter 0]

9 Jun 2025

PONE-D-25-26679Short-term power prediction of photovoltaic power stations based on Kepler algorithm and VMD-CNN-LSTM modelPLOS ONE

Dear Dr. wang,

Thank you for submitting your manuscript to PLOS ONE. After careful consideration, we feel that it has merit but does not fully meet PLOS ONE’s publication criteria as it currently stands. Therefore, we invite you to submit a revised version of the manuscript that addresses the points raised during the review process.

We look forward to receiving your revised manuscript.

Kind regards,

Ardashir Mohammadzadeh, Phd

Academic Editor

PLOS ONE

6. PLOS requires an ORCID iD for the corresponding author in Editorial Manager on papers submitted after December 6th, 2016. Please ensure that you have an ORCID iD and that it is validated in Editorial Manager. To do this, go to ‘Update my Information’ (in the upper left-hand corner of the main menu), and click on the Fetch/Validate link next to the ORCID field. This will take you to the ORCID site and allow you to create a new iD or authenticate a pre-existing iD in Editorial Manager.

Reviewers' comments:

Reviewer's Responses to Questions

**Comments to the Author**

1. Is the manuscript technically sound, and do the data support the conclusions?

Reviewer #1: Yes

Reviewer #2: Partly

2. Has the statistical analysis been performed appropriately and rigorously? 

Reviewer #1: Yes

Reviewer #2: Yes

3. Have the authors made all data underlying the findings in their manuscript fully available?

Reviewer #1: Yes

Reviewer #2: Yes

4. Is the manuscript presented in an intelligible fashion and written in standard English?

Reviewer #1: Yes

Reviewer #2: Yes

5. Review Comments to the Author

Reviewer #1: The research addresses the short-term power prediction of photovoltaic power stations, aiming to tackle the intermittent and fluctuating nature of photovoltaic power generation. The study introduces the Kepler algorithm to analyze historical data deeply, uncovering nonlinear relationships among various factors. This lays a solid foundation for subsequent predictions. A combined model of VMD (Variational Mode Decomposition), CNN (Convolutional Neural Network), and LSTM (Long Short-Term Memory) is developed. VMD decomposes the original power sequence based on frequency characteristics, which helps reduce data complexity

The manuscript has a solid foundation and offers valuable insights into

PV system applications. However, improvements in writing

clarity, methodological transparency, and critical discussion are necessary

before it is suitable for publication.

1. The title can be modified to avoid using abbreviations, which can be more effective academically.

2. If you use abbreviation, it's often helpful to spell it out in the abstract to clarify meaning for readers and reviewers. Some abbreviations are not declared, such as VMD-CNN-LSTM and KTH. As reviewer I understand sometimes if a common abbreviation in the field no need to be declared, but assume a reader is reading this without pervious knowledge or background.

3. The abbreviation needs to be clear such as Kepler Optimization Algorithm (KOA) and not misleading the reader.

4. Abbreviations need to be declared in the first time mentioned in the text, as an example Long Short-Term Memory (LSTM) is declared in section III, while it has been mentioned before in different sections.

5. In section 4.1, the word chapter has been highlighted. It is better to replace with word section to keep the academic consistency.

6. The performance of the model relies heavily on the quality and quantity of historical data. Poor or missing historical data can lead to poor predictions. The paper does not explore how varying levels of data quality can influence model performance.

7. The performance of the model is heavily reliant on the quality and quantity of historical data. If the historical data is not comprehensive or contains factual inaccuracies, then it could lead to worse predictions. The paper does not discuss how variations in the quality of data may impact the performance of the model.

8. The model's performance is highly reliant on the quantity and quality of historical data. If historical data lacks completeness or accuracy, it could lead to poor predictions. How variations in data quality can impact the model's performance is not covered in the research.

9. The performance of the model is highly dependent on the quantity and quality of historical data. If historical data is not complete or is inaccurate, it may result in suboptimal

10. In addition, for more insight to the work. These key points need to clear to give strength weight academically. How does Kepler Algorithm enhance prediction accuracy? What data sources were used for historical analysis? How does VMD impact data complexity reduction?

Reviewer #2: This paper presents an approach to the problem of short-term power prediction of photovoltaic power stations using the Kepler Optimization Algorithm (KOA) combined with Variational Mode Decomposition – Convolutional Neural Network – Long Short-Term Memory models.

The authors should consider the following issues:

- Page 1 and further: the authors refer to the KOA as inspired by Kepler’s laws of planetary motion. Considering that the same source of inspiration is used by the Gravitational Search Algorithm (GSA), the authors should refer also to GSA in their review. A brief comparison between KOA and GSA should b added.

- Pages 2 and 3, subsection 2.1: in Step 1, the authors refer ti i and N as “planets”, while further they refer the same parameters as being “objects”. Moreover, further away, in Step 3, they use the term “target planet”. Please, use a unique term.

- Page 3, eq. (3) as compared to other equations: explain the difference between parameter “r” and “rand[0,1]”.

- Page 3, text after eq. (4):chek notation M_S.

- Page 3: explain the rational of formula in eq. (6).

- Page 3, eq. (6) and text after it: do not use small and capital letters for the same variable/parameter: see v cs. V, x vs. X etc

6. PLOS authors have the option to publish the peer review history of their article (what does this mean? ). If published, this will include your full peer review and any attached files.

**Do you want your identity to be public for this peer review?** For information about this choice, including consent withdrawal, please see our Privacy Policy .

Reviewer #1: No

Reviewer #2: No

---

## [Author Response · Author response to Decision Letter 1]

7 Jul 2025

We are uploading (a) a rebuttal letter that responds to each point raised by the academic editor and reviewer(s). (b)a marked-up copy of your manuscript that highlights changes made to the original version. and (c) an unmarked version of your revised paper without tracked changes.

---

## [Decision Letter · Decision Letter 1]

22 Jul 2025

基于开普勒优化算法和 VMD-CNN-LSTM模型的光伏电站短期功率预测

PONE-D-25-26679R1

Dear Dr. wang,

We’re pleased to inform you that your manuscript has been judged scientifically suitable for publication and will be formally accepted for publication once it meets all outstanding technical requirements.

Kind regards,

Ardashir Mohammadzadeh, Phd

Academic Editor

PLOS ONE

Additional Editor Comments (optional):

Reviewers' comments:

Reviewer's Responses to Questions

**Comments to the Author**

1. If the authors have adequately addressed your comments raised in a previous round of review and you feel that this manuscript is now acceptable for publication, you may indicate that here to bypass the “Comments to the Author” section, enter your conflict of interest statement in the “Confidential to Editor” section, and submit your "Accept" recommendation.

Reviewer #1: All comments have been addressed

Reviewer #2: All comments have been addressed

2. Is the manuscript technically sound, and do the data support the conclusions?

Reviewer #1: Yes

Reviewer #2: Yes

3. Has the statistical analysis been performed appropriately and rigorously? 

Reviewer #1: (No Response)

Reviewer #2: Yes

4. Have the authors made all data underlying the findings in their manuscript fully available?

Reviewer #1: Yes

Reviewer #2: Yes

5. Is the manuscript presented in an intelligible fashion and written in standard English?

Reviewer #1: Yes

Reviewer #2: Yes

6. Review Comments to the Author

Reviewer #1: Thank you for considering the reviewers' comments, which have helped sharpen the research and improve its academic quality.

Reviewer #2: All comments have been addressed by the authors. The quality of the paper has been enhanced.

No more comments

7. PLOS authors have the option to publish the peer review history of their article (what does this mean? ). If published, this will include your full peer review and any attached files.

**Do you want your identity to be public for this peer review?** For information about this choice, including consent withdrawal, please see our Privacy Policy .

Reviewer #1: No

Reviewer #2: No

---

## [Editor Report · Acceptance letter]

PONE-D-25-26679R1

PLOS ONE

Dear Dr. wang,

I'm pleased to inform you that your manuscript has been deemed suitable for publication in PLOS ONE. Congratulations! Your manuscript is now being handed over to our production team.

Kind regards,

on behalf of

Dr. Ardashir Mohammadzadeh

Academic Editor

PLOS ONE